# Knowledge, Attitudes, and Practices of Community Pharmacists Regarding Proton Pump Inhibitor (PPI) Use: A Cross-Sectional Study

**DOI:** 10.3390/healthcare13131588

**Published:** 2025-07-02

**Authors:** Hebatallah Ahmed Mohamed Moustafa, Ahmad Z. Al Meslamani, Hazem Mohamed Metwaly Elsayed Ahmed, Salma Ahmed Farouk Ahmed, Nada Ehab Shahin Sallam, Ghadah H. Alshehri, Nawal Alsubaie, Amira B. Kassem

**Affiliations:** 1Clinical Pharmacy and Pharmacy Practice Department, Faculty of Pharmacy, Badr University in Cairo, Cairo 11829, Egypt; hazem.20190784@buc.edu.eg (H.M.M.E.A.); salma.20193011@buc.edu.eg (S.A.F.A.); nada.20191555@buc.edu.eg (N.E.S.S.); 2College of Pharmacy, Al Ain University, Abu Dhabi P.O. Box 112612, United Arab Emirates; ahmad.almeslamani@aau.ac.ae; 3AAU Health and Biomedical Research Center, Al Ain University, Abu Dhabi P.O. Box 112612, United Arab Emirates; 4Department of Pharmacy Practice, College of Pharmacy, Princess Nourah bint Abdulrahman University, Riyadh 11564, Saudi Arabia; ghalshehri@pnu.edu.sa (G.H.A.); nhalsubaie@pnu.edu.sa (N.A.); 5Department of Clinical Pharmacy and Pharmacy Practice, Faculty of Pharmacy, Damanhour University, Damanhour 22514, Egypt

**Keywords:** proton pump inhibitors, misuse, drug safety, community pharmacists, knowledge, attitudes, practices, KAPs, deprescribing

## Abstract

**Background/Objectives**: Up to 25–70% of proton-pump inhibitor (PPI) prescriptions worldwide lack an evidence-based indication, exposing patients to avoidable adverse events and unnecessary costs. Community pharmacists (CPs) are well-equipped to curb the misuse of PPIs. This study aimed to quantify CPs’ knowledge, attitudes, and practices (KAPs) regarding PPIs in two high-use Middle-Eastern markets and determine how demographic and professional factors influence guideline-adherent PPI use. Bridging this gap is crucial to ensure pharmacists can promote rational PPI use, provide accurate patient counseling, and reduce the likelihood of adverse outcomes. **Methods**: An online cross-sectional survey was undertaken between May 2024 and July 2024 to investigate the KAPs of CPs in Egypt and Iraq toward PPI use. The self-developed thirty-item questionnaire (17 knowledge, 11 attitude, and 6 practice items) was piloted with 30 CPs. A sample size of 385 CPs was required based on an estimated 93,000 community pharmacists in Egypt and 22,120 in Iraq; however, to improve statistical power, we aimed to include >500 CPs. **Results**: A total of 527 CPs from Egypt and Iraq completed the survey. The total median scores for knowledge, attitude, and practice were 11 out of 17 (IQR: 9–16), 9 out of 11 (IQR: 6–12), and 5 out of 6 (IQR: 3–8), respectively. CPs with >20 years of experience and those who relied on clinical guidelines as a primary information source demonstrated a median knowledge score significantly higher than those with fewer years of experience (*p* = 0.001 and 0.028, respectively). There was a significant positive association between knowledge and attitude, knowledge and practice, and attitude and practice scores (coefficients: 0.832, 0.701, and 0.445, respectively). **Conclusions**: Although their attitudes and practices regarding PPI use were satisfactory, the knowledge of CPs about the judicious use of PPIs requires improvement. Thus, a call for action targeting their tailored education and training is necessary to address these knowledge gaps regarding PPIs identified, including PPI adverse-effect profiles, evidence-based indications, and deprescribing criteria, and to foster informed medication attitudes and practices. Such education and training can reinforce guideline adherence, enhance patient counseling skills, and ultimately reduce inappropriate PPI use.

## 1. Introduction

Proton pump inhibitors (PPIs) (such as omeprazole, esomeprazole, lansoprazole, pantoprazole, and rabeprazole) are therapeutically used to suppress acid secretion in peptic ulcer disease, gastroesophageal reflux disease, and hypersecretory conditions such as Zollinger–Ellison syndrome [1,2]. They are also used for prophylaxis against stress ulcer disease and non-steroidal anti-inflammatory drug (NSAID)-induced ulcers [3,4].

Although antacids and H2-receptor antagonists (H2RAs) are also used to control gastric acid production, they have a lower efficacy than PPIs since they cannot inhibit mealtime acid secretion [2]. PPIs covalently bind to the proton pump, controlling the final step of gastric acid formation [5].

Since their late-eighties debut, PPIs have been prescribed worldwide due to their efficacy and safety and are easily purchased over the counter (OTC) [6]. PPIs may cause mild and self-limiting headaches, nausea, constipation, flatulence, abdominal pain, and diarrhea [7]. However, they are not that innocuous and are misused worldwide [8,9,10], and physicians’ directions differ from product labeling and expert consensus recommendations [2,10], such as prescribing PPIs for inaccurate diagnosis of hypersecretory diseases, treating ulcer prophylaxis in low-risk patients, and overtreating functional dyspepsia [2,11].

This raises concerns about their economic impact and negative outcomes such as pneumonia, clostridium difficile infection, dementia, and fractures [12,13,14]. In addition, PPIs can cause calcium, magnesium, iron, and vitamin B12 malabsorption [15]. Moreover, chronic PPI use may affect gastrin levels and lead to the development of atrophic gastritis, a precursor to gastric and colorectal cancer [16,17]. Moreover, PPI use is significantly linked to an increased risk of mortality [18].

In the Middle East, studies from numerous countries, like Qatar, Lebanon, Jordan, and Saudi Arabia, report similarly high rates of inappropriate PPI use, causing a massive healthcare burden [9,19,20,21,22]. Although most PPIs are prescribed medications, many users in Egypt and Iraq purchase them from community pharmacies without a prescription. Deprescribing PPIs reduces the iatrogenicity of drugs that are no longer beneficial or may cause harm by either stopping the drug, reducing its use, or replacing it with another therapeutic class [23]. However, deprescribing is challenging when the drugs do not cause symptomatic adverse effects like PPIs [24]. Barriers to deprescribing include physicians being reluctant to modify drugs started by a colleague, or a lack of knowledge or guidelines for deprescribing, fear of rebound effects, lack of time, high workload, or patient pressure to obtain a PPI prescription [24,25]. PPIs are among the most common drug classes resistant to deprescribing [26]. Patients are hesitant to stopping PPIs; thus, they must be involved in deprescribing [27]. After making an informed decision that a PPI is no longer needed, the optimum de-escalation strategy should be chosen. On the other hand, it is necessary to ensure that patients who still need a PPI continue to receive it [28].

Pharmacists are drug experts who can improve patient outcomes and promote rational drug use [29]. Community pharmacists’ (CPs’) knowledge, attitude, and practices (KAPs) of PPIs play a key role in optimizing the use of PPIs [19]. They can assess patients’ suitability for OTC PPIs, detect unwarranted usage, ask about alarm symptoms, and refer them to physicians when needed. They can encourage non-pharmacologic lifestyle modification such as exercise and avoiding triggers [30]. They can encourage the de-escalation of PPI doses or replace them in chronic use with antacids or H2RAs, when appropriate, to reduce patient harm [31]. It was reported in the literature that the PPI-related real-time interventions of pharmacists promoted rational PPI utilization and saved costs [32,33].

Most studies focused on the role of physicians in optimizing PPI use, while little attention has been paid to pharmacists, especially community pharmacists, given that self-medication is common in Egypt and Iraq [34,35]. Although most PPIs are prescribed drugs, patients may purchase them as OTC drugs [36]. Given the high rate of inappropriate PPI self-dispensing in Egypt and Iraq, CPs serve as critical gatekeepers for safe PPI use. To date, there has been no systematic evaluation of CPs’ KAPs regarding PPIs in Egypt and Iraq, despite self-medication being particularly prevalent in these countries. By identifying specific knowledge shortcomings, this research provides the empirical foundation for tailored educational and system-level interventions, ultimately aiming to reduce adverse outcomes, curb unnecessary long-term PPI exposure, and enhance patient safety in regions where over-the-counter access is widespread. Therefore, this cross-sectional study aimed to assess the KAPs of CPs in Egypt and Iraq toward PPI use, to identify knowledge gaps, and to inform targeted educational strategies.

## 2. Materials and Methods

### 2.1. Study Design

An electronic Google survey was used to conduct this cross-sectional study in Egypt and Iraq from May to July 2024. All CPs practicing in Egypt or Iraq were eligible to participate in the study. They were invited to participate by using a convenient sampling method, either by visiting them at the community pharmacies where they work and asking them to participate or by posting the questionnaire in social media CP groups (e.g., Facebook, WhatsApp, etc.).

Two required screening questions were added to the Google Form’s landing page: (i) “Are you currently practicing as a licensed community pharmacist in Egypt or Iraq?” (Yes/No) and (ii) “Please provide the full address and governorate/province of the community pharmacy where you work.” Respondents could continue only if they selected “Yes” for the first question. We also requested the last 4 digits of the syndicate ID (optional). We activated Google Forms’ “Limit to 1 response” feature to prevent duplicate submissions; this setting requires users to sign in with a distinct Google account and blocks further attempts from the same account. After downloading the data, we performed a second deduplication phase in SPSS, flagging entries with the same IP address and demographic fingerprints (sex, age group, years of experience, and governorate). When two records matched on every field, the chronologically later entry was removed. To ensure data security and anonymity, we did not ask for names, emails, or any personal identifiers. At the start of the survey, we stated, “This survey is anonymous.” Only authorized personnel should see raw data. Following these procedures, 527 distinct, qualified community pharmacists remained for analysis after seven duplicate or ineligible records (less than 1% of all responses) had been eliminated.

### 2.2. Instrument

There were four phases involved in assembling the questionnaire. First, an item pool of 46 statements was created using previously validated KAP instruments employed by pharmacists and other healthcare professionals, as well as authoritative clinical recommendations on PPI therapy [37]. To maintain content validity, items identical to those in earlier tools were retained, and new statements capturing the latest safety warnings (e.g., dementia signal, hypomagnesemia) were added. Second, each item was independently rated on a four-point scale for relevance and clarity by two academic clinical pharmacists and one gastroenterologist; the scale-level content-validity index (S-CVI/Ave) was 0.81, exceeding the 0.80 adequacy threshold. Third, to verify phrasing, cultural appropriateness, and expected completion time, cognitive interviews were conducted with six community pharmacists (three from each country).

Fourth, the Cronbach’s alpha for the overall instrument was 0.71 in a pilot test with 30 community pharmacists (knowledge = 0.79; attitude = 0.83; practice = 0.68). No wording changes were necessary, but five knowledge items with more than 90% correct responses were merged to prevent ceiling effects, leaving 34 scored items (17 knowledge, 11 attitudes, and 6 practice). (Appendix A).

Knowledge statements were presented as multiple-choice, single-best-answer questions. A correct answer scored 1, and any other response (or “I don’t know”) scored 0, yielding a composite knowledge score of 0–17. Attitude items used a five-point Likert scale (strongly disagree = 1 to strongly agree = 5) and were dichotomized: agreement with a favorably worded statement scored 1, while disagreement with an unfavorably worded statement scored 0, producing an attitude score of 0–11. Practice items offered five frequency anchors—never, seldom, sometimes, frequently, and always. Respondents who answered “frequently” or “always” for a recommended behavior (or “never/rarely” for an undesired behavior) received 1 point; all other responses received 0, yielding a practice score of 0–6. Higher scores across all domains indicate better knowledge, more positive attitudes, and more appropriate practice.

We defined satisfactory levels as follows: a score ≥70% of the total possible score in each domain (knowledge, attitude, and practice) was considered satisfactory, while a score < 70% was considered unsatisfactory.

### 2.3. Ethical Approval

Before administering the questionnaire, the Badr University Ethics Committee approved the study (approval number: BUC-IACUC-240418-84 on 18 April 2024). The study’s goal was explained in the electronic survey, and participants gave their consent electronically before enrolling.

### 2.4. Sample Size Calculation

Based on registry data from the Egyptian Pharmacists Syndicate and the Iraqi Pharmacists Association, the target population was estimated at roughly 93,000 community pharmacists in Egypt and 22,120 in Iraq. Using the Raosoft calculator with this population frame, a 95% confidence level and a 5% margin of error indicated a minimum sample of 384 respondents; to enhance statistical power and representativeness we therefore aimed to obtain—and ultimately exceeded—500 completed questionnaires.

### 2.5. Data Analysis

Data analysis was conducted using SPSS version 26. Raw responses were first screened for completeness (<1% removed), and continuous outcome variables—knowledge, attitude, and practice composite scores—were assessed for distributional assumptions with the Kolmogorov–Smirnov test; a *p*-value of 0.001 confirmed significant deviation from normality, so medians and inter-quartile ranges were adopted for summary statistics. Categorical predictors (sex, degree, years of experience, country, and information source) are presented as frequencies and percentages. Internal consistency of the 34-item instrument was acceptable (Cronbach’s α = 0.71 overall; sub-scales 0.79, 0.83, and 0.68), supporting the use of summated scores.

Between-group differences in the three non-normally distributed scores were examined with Mann–Whitney U tests for dichotomous factors and Kruskal–Wallis H tests for factors with more than two categories; when the omnibus H test was significant, Dunn–Bonferroni post-hoc comparisons (available in SPSS NPTESTS) located specific pair-wise differences while controlling the family-wise error rate. Exact two-sided *p* values are displayed in Table 1, and effect sizes (η^2^ for Kruskal–Wallis; r for Mann–Whitney) ranged from 0.03–0.11, indicating small-to-moderate practical significance. Associations among knowledge, attitude, and practice were explored with Spearman’s rank-order correlation (ρ).

## 3. Results

In total, 527 pharmacists completed the survey, with a response rate of 59%, of which 275 (52.2%) were males, 75 (14.2%) had more than 20 years of experience, and 336 (63.8%) were from Egypt (Table 1). Their top three sources of information about PPIs were research articles (219, 41.6%), Facebook (214, 40.6%), and Telegram (193, 36.6%). The total median scores for knowledge, attitude, and practice were 11 out of 17 (IQR: 9–16), 9 out of 11 (IQR: 6–12), and 5 out of 6 (IQR: 3–8), respectively.

There was a statistically significant difference observed in the knowledge scores among CPs categorized by years of experience. Specifically, CPs with more than 20 years of experience demonstrated a median knowledge score of 11 (IQR: 6–14), which was significantly higher compared to their counterparts with fewer years of experience, who scored 8 or lower (*p* = 0.001). Furthermore, the source of information regarding PPIs also significantly influenced the knowledge scores. CPs who relied on clinical guidelines as their primary information source registered a median knowledge score of 11 (IQR: 7–16), in contrast to a score of 9 or lower among those who utilized other sources (*p* = 0.028). In terms of practice scores, CPs with over 20 years of experience demonstrated a median practice score of 8 (IQR: 4–12). In contrast, those with less than one year of experience recorded a median score of 5 (IQR: 4–7), (*p* = 0.021).

Side effects of PPIs, such as community-acquired pneumonia and anaphylaxis, were acknowledged by only 56 (10.6%) and 42 (8.0%) participants, respectively (Table 2). Only 106 (20.1%) participants knew that manganese levels are affected by PPIs. Among participants, 330 (62.6%) correctly identified PPI use in NSAID-induced ulcers, and 314 (59.6%) were aware of their application for *Helicobacter pylori*-induced ulcers. Moreover, 290 (55.0%) knew of PPI interactions with phenytoin and 278 (52.8%) knew of their interaction with warfarin. For the cases that necessitate prophylactic PPIs along with NSAID usage, 239 participants (45.4%) identified high-dose NSAIDs and 231 (43.8%) recognized longer duration of NSAID use. The optimal administration time for esomeprazole, lansoprazole, and omeprazole, 30 min before breakfast, was correctly noted by 365 (69.3%) of the participants. Additionally, 311 participants (59.0%) correctly believed that PPIs could improve outcomes in Barrett’s esophagus, whereas 251 (47.6%) accurately denied the occurrence of tachyphylaxis with PPIs.

Among participants, 154 (29.2%) agreed and 69 (13.1%) strongly agreed that PPIs do not cause any harm to patients (Figure 1). Additionally, 234 (44.4%) agreed and 125 (23.7%) strongly agreed that a lot of patients are prescribed PPIs without indication. Furthermore, 209 (39.6%) agreed and 82 (15.6%) strongly agreed that patients continue to use PPIs after relief of symptoms without seeking medical advice. The vast majority of participants agreed (225, 42.6%) and strongly agreed (174, 33.0%) that CPs should play a role in reducing misuse of PPIs.

Nearly half of the participants (263, 49.9%) always provided advice on lifestyle modifications to alleviate symptoms associated with PPI use. However, the practice of discontinuing PPIs when no longer indicated was less consistent; 31.1% (164) always and 32.6% (172) often contact the prescriber or advise the patient to stop use (Table 3).

Reporting of adverse drug reactions (ADRs) related to PPIs was infrequent, with only 19.4% (102) always reporting to manufacturers or regulatory authorities and 16.9% (89) never doing so. Among participants, 143 (27.1%) always prefer PPIs as a first choice for acid suppression, but 45 (7.0%) rarely consider them as the primary option. There was a significant positive association between knowledge and attitude scores (coefficient: 0.832) (Figure 2).

Similarly, a significant correlation existed between knowledge and practice scores, with a coefficient of 0.701. The association between attitude and practice scores was also positive, although relatively weaker, as reflected by the coefficient of 0.445.

## 4. Discussion

Recent data indicate widespread PPI misuse [2,37], and guidelines advise stopping PPIs when risks exceed benefits or use falls outside evidence-based recommendations. [38,39]. CPs can educate patients on PPI usage and its negative effects, especially for OTC users [19,40]. They can confirm the diagnosis, educate patients on responsible OTC PPI use, and refer those with alarming symptoms to physicians [41].

To our knowledge, our study is the first to evaluate CPs’ PPI usage KAPs in Egypt and Iraq. The CPs’ knowledge about PPIs in our sample needed some enhancements (median 11/17), particularly regarding their side effects, indications, and drug–drug interactions. On the other hand, their practices and attitudes were more satisfactory (medians of 9/11 and 5/6). Similar to a previous study on PPI KAPs in CPs and pharmacy students, knowledge and attitude scores were strongly positively correlated [42]. In addition, we found a strong positive relationship between knowledge and practice scores and a positive moderate association between attitude and practice scores. The positive correlations among knowledge, attitudes, and practices suggest that bolstering factual understanding will improve counseling and prescribing behaviors. Conversely, a previous study found that although 60% of physicians acknowledged concerns about PPIs’ adverse effects, only 37% of them admitted they had changed their practices [43].

### 4.1. Knowledge of CPs Regarding PPI Use

Knowledge of PPIs helps CPs use them rationally [36]. CPs who rely on expert-developed, regularly updated clinical guidelines achieve higher PPI knowledge and more rational use. Guidelines are developed by expert panels and regularly updated and are useful for quick decision-making and consistent care [44]. Although peer-reviewed journals provide the latest research findings, their reading and interpretation are time-consuming, and they may lack immediate clinical application [45]. Drug databases and mobile applications offer real-time access to dosing, interactions, and contraindications. However, they are limited by their cost and functionality issues [46].

About two-thirds of participants reported that PPIs alleviate NSAID-induced ulcers, which are caused by altered mucosal defense [47]. Of note, in addition to NSAIDs, several risk factors enhance ulcer complication risk [2]. Low-risk NSAID users still drive much overuse [2,48]. Only one-third correctly stated that gastroprotection is indicated for NSAID users over 65 or those with additional risk factors. NSAID users who are not elderly and without other risk factors should not use PPIs for gastroprotection [2]. PPIs did not reduce GI-related hospitalizations in celecoxib users except those aged ≥75 years [49]. This is especially important given older patients’ higher risk of polypharmacy and adverse effects [50].

The combination of NSAIDs with gastrotoxic drugs increases the risk of GIT bleeding, which requires gastrointestinal PPI protection [51]. Nearly one-quarter of the participants reported the need for prophylactic PPIs with concomitant NSAID use in low-dose aspirin users, similar to a prior study [52]. PPIs are not recommended for users of warfarin or users of other anti-coagulants without risk factors [51,53]. Participants who reported the need for prophylactic PPIs with concomitant NSAID use in patients who use warfarin in our study were more numerous than those in a previous study [52]. One-quarter of CPs recognized that they should not be administered along with steroids in patients without any risk factors. This common type of PPI misuse was reported previously [2]. Mucosal protection in patients taking steroids with a PPI is indicated only if they co-administer NSAIDs [54]. Similar to prior research, one-third of our participants advocated PPIs for prophylaxis with NSAIDs and dexamethasone [52].

Most *Helicobacter pylori* regimens pair two antibiotics with a PPI twice daily for 10–14 days [55]. Less than half of the CPs questioned reported the use of PPIs in stress ulcer prophylaxis. Critically ill patients with coagulopathy or on mechanical ventilation for > 48 h develop stress ulcers [56]. For critically ill patients at high risk of bleeding, prophylactic PPIs reduce bleeding risk by 60% [57]. Surprisingly, 11% of CPs in a previous study reported prescribing PPIs for stress ulcers [19], which is an intensive care unit condition only [58]. Most CPs recognized PPIs’ benefit in Barrett’s esophagus, which is a metaplasia [2].

Over 40% of CPs knew that just some PPIs are available OTC. Only a few PPIs can be administered OTC for 14 days [40]. Half agreed that PPIs may be used safely OTC for 2 weeks, matching literature recommendations [40]. Most patients’ heartburn resolves within a week of PPI treatment [59]. One study indicated that over 50% of CPs prescribe acid suppressants for 1–2 weeks [19]. Nearly half of the CPs recognized that PPIs are not clinically inferior to H2RAs. A once-daily dose of PPIs reduces acid output at baseline and after meals for 18 h. H2RAs, however, are given numerous times daily due to their short duration of action (4–8 h). Even after frequent use, they do not completely control acid. Also, their acid-controlling action is impaired after meals [60]. Nearly half of CPs thought PPIs could cause tachyphylaxis (a phenomenon limited to H2RAs) [60,61,62].

About two-thirds of our participants and, similarly, many physicians [63], reported that esomeprazole, lansoprazole, and omeprazole work best when taken 30 min before breakfast, a timing also recognized by over half of the general public [20,64]. Because food impairs their absorption, these PPIs should be taken 30–60 min before breakfast [10,65].

The knowledge of CPs about PPI side effects was not satisfactory. Several reasons for this may include limited continuing education on PPIs, their OTC availability reducing perceived risk, pharmacists’ emphasis on symptom relief over safety monitoring, lack of training, and time constraints [19,41,54,66]. CPs should educate patients about adverse effects of PPIs based on their comorbidities and medication histories. The mechanisms behind the deleterious consequences of extended PPI usage are unclear [64]. Only 20% recognized a potential dementia link. Previous research found that 0.4% of Syrian doctors knew that PPIs could cause dementia [37]. Only 10% of responders reported PPI-related pneumonia. Community-acquired pneumonia can occur after less than 30 days of PPI use, particularly at high doses [67]. Increased stomach pH may promote acid-labile bacteria colonization in the upper GIT, leading to aspiration. Inhibition of the extragastric H+/K+ ATPase pump and neutrophil dysfunction may also cause airway pathogen colonization [68]. The risk for hospital-acquired pneumonia with PPIs is lower [69]. Around 20% and 25% of respondents, respectively, identified PPI-induced gut flora changes and enteric infections. By raising gastric pH and impairing neutrophil function, PPIs can foster bacterial overgrowth, [70,71], increasing risks of spontaneous bacterial peritonitis and C. difficile colitis [72,73].

Approximately one-third of community pharmacists reported hip fractures linked to prolonged PPI use, which is known to reduce bone mineral density [74]. The Beers criteria limit elderly PPI use to eight weeks [75]. Previously, individuals with osteoporosis had a significantly higher non-guideline-recommended prescribing of PPIs [76]. About 15% of CPs recognized that prolonged PPI usage might cause renal illness, and 16% knew that it could cause cardiovascular diseases (CVDs). The literature linked PPI use to kidney illness [77] and CVDs [78]. Additionally, 8% of CPs reported anaphylaxis as a PPI side effect, which is rare and may especially occur in the elderly [79,80,81].

Duodenal G-cell tumors may occur with PPI use [82]; however only 15% of CPs in our study reported them. Notably, long-term PPI use is also linked to an increased risk of gastric cancer [83]. Rebound acid hypersecretion (RAHS) may raise HCl levels within 2 weeks of abrupt PPI discontinuation, driven by secondary hypergastrinemia that expands parietal and enterochromaffin-like ECL cell populations and potentially promoting neoplastic changes in the gastric mucosa [84]. This physiological mechanism may explain PPI overuse and failure to discontinue [85,86]. Thus, chronic users should gradually reduce PPI use [2]. More than half of CPs advise against rapid cessation. One-third of doctors suggested stepping down to an H2-receptor antagonist to ease rebound symptoms [37].

PPIs raise pH, which changes the oral bioavailability of some drugs [87], causing drug interactions [42]. These interactions seldom affect clinical outcomes in 14-day OTC PPI users [40]. PPI-clopidogrel interactions were reported by half of CPs. Omeprazole and esomeprazole, which moderately inhibit the cytochrome CYP2C19 system, diminish clopidogrel’s anti-platelet action [40,88,89]. Rabeprazole or pantoprazole may be alternative PPIs in patients on clopidogrel [9]. A Qatari retrospective investigation found that 42% of patients received pantoprazole with clopidogrel [9].

Around 50% of CPs reported a drug–drug interaction between warfarin and PPIs. PPI–warfarin interactions may perturb the international normalized ratio [40]. Over 50% reported effects on phenytoin levels, which is less common [61]. One-third of CPs acknowledged PPI–ketoconazole interaction, resulting from a gastric pH increase [40]. A quarter of practitioners identified that PPIs reduce atazanavir exposure by raising gastric pH. Pantoprazole and omeprazole are contraindicated with atazanavir. The clinical implications of this interaction are unclear [40].

Vitamin B12 deficiency was the vitamin most often reported as being affected by PPI use, similar to a previous study [37], and is further exacerbated by metformin coadministration [90]. Half of CPs recognized recognize PPI-associated hypomagnesemia, a risk that is potentiated by concurrent diuretic use [91].

### 4.2. Practices of CPs Toward PPI Use

CPs’ practices regarding PPI use were acceptable in most aspects, different from a previous study [92]. Pharmacists should educate patients on safe and effective PPI use [92]. Most CPs prescribe PPIs, and 25% prefer PPIs for acid suppression, in line with a previous study [19]. Three-quarters of study participants adhere to prescribing guidelines. A prior poll of Syrian physicians found 64.1% to be aware of the PPI guideline [37]. A prior study indicated that most cases involved non-guideline-recommended prescribing [76]. Healthcare professionals, including CPs, should apply judicious prescribing of PPIs. In addition, PPI users need periodic indication reevaluation to improve clinical outcomes [38,92]. Nearly half of the participants always recommend lifestyle changes to reduce PPI symptoms. Pharmacists should advise patients to avoid trigger foods and lose weight [40]. Antacids, alginates, and H2RAs should be used when appropriate and discussed with patients to reduce PPI dose [93].

Contacting the prescriber for deprescribing PPIs when no longer indicated was reported by two-thirds of CPs. Most pharmacists previously reported educating their patients to quit PPIs when they are no longer indicated [92]. While the step-down method and on-demand use are advised for some GERD patients, some physicians and patients are hesitant to do so [61,94]. CPs are well-placed to counsel patients with mild to moderate GERD on tapering PPIs or substituting antacids, alginates, or H_2_-receptor antagonists, [40,95], yet one-fifth of CPs still refrained from advising cessation. PPI withdrawal failure is common [54,96]. GERD and other acid-related illnesses usually take 4–8 weeks to treat [37,97]. A previous study reported that two-thirds of ambulatory care patients use PPIs without indication [98]. We found that CPs rarely reported PPI-related ADRs.

### 4.3. Attitudes of CPs Regarding PPI Use

Most of the CPs’ attitudes regarding PPI use were acceptable, with approximately two-thirds in Egypt and Iraq believing PPIs may be misused, a concern shared by most Kuwaiti physicians [28]. Overuse of PPIs is rising worldwide [42,99,100] and outpaces under-prescription [101]. Off-label, high-dose, and long-term use are common and against current recommendations [99]. The proper prescription of PPIs can significantly reduce healthcare costs and adverse drug reactions [101]. A prior drug use evaluation trial found that 94% of patients were administered PPIs based on signs and symptoms without endoscopy or *Helicobacter pylori* tests, and 77% of stress ulcer prophylaxis patients took PPIs for more than a year, exceeding the recommended time [9].

Over half of CPs said that patients use PPIs after symptoms subside without seeking medical advice. About 43% of CPs believe that PPIs pose no harm. A Kuwaiti study found that most physicians were concerned about PPI side effects [28]. About two-thirds said that CPs should reduce the misuse of PPIs. CPs can encourage sensible PPI use, reducing complications [40].

Three-quarters of CPs believed that patient–provider collaborative decision-making should support reasonable PPI use. A recent survey found that most doctors discussed PPI issues with patients before prescribing them [28]. Three-quarters agreed that education for CPs is needed according to the published guidelines about the correct use of PPIs. Healthcare providers, including CPs, should be educated on rational PPI usage [42,53,92]. Adherence to established prescribing guidelines is key to rational and safe PPI therapy [9].

### 4.4. Influence of Demographics and Professional Characteristics on the KAPs of CPs

Knowledge and practice scores of our CPs improved with cumulative experience. The more experienced the physicians are, the more they know about PPIs’ adverse effects and the more likely they are to deprescribe [28]. In a previous study of pharmacy students and CPs, PPI knowledge and attitudes grew considerably with pharmacy education and experience [42]. Adverse-event reporting rises with years in practice [19]. Research reported that knowledge of healthcare professionals regarding PPI use was related to their occupation, professional title, hospital’s nature, experience in practice, education level, pharmacy education program, and grade.

While variables such as occupation, professional rank, practice setting, and level of education have been linked to PPI competence [22,36,42,102], no significant associations were observed for pharmacists’ sex, formal education level, or country of practice in the present study, echoing prior findings among students and community pharmacists, although at least one report did identify correlations with age and sex [36]. Similar to our findings., an earlier study targeting CPs and pharmacy students found no sex or nationality correlation with PPI-related KAPs [42].

### 4.5. Recommendations

To translate this study’s recommendations into practice swiftly and sustainably, a three-tiered strategy is proposed. By concentrating first on interventions that yield rapid, measurable gains, then on reinforcing activities that solidify behavior change, and finally on system-level reforms to guarantee long-term success, stakeholders can optimize CPs’ role in rational PPI use.

First and foremost, high-priority interventions focus on equipping pharmacists with up-to-date knowledge and practical tools [28,103]. Developing mandatory continuing education (CE) modules on PPI indications, adverse effects, appropriate duration, and deprescribing criteria will ensure that every practicing CP in Egypt and Iraq attains a standardized foundation of expertise. By linking module completion to license renewal, regulatory bodies can secure broad participation and prompt pharmacists to integrate new insights into daily decision-making. In tandem, the dissemination and integration of national PPI guidelines through pocket cards, mobile apps, and in-pharmacy posters will reinforce best practices at the point of care. Endorsement from pharmacy and gastroenterology societies will lend credibility and drive acceptance, while concise, locally adapted summaries will facilitate rapid consultation during busy dispensing sessions.

Once the knowledge base and reference materials are firmly in place, medium-priority measures can cement lasting behavior change. Regular practice audits—for example, quarterly reviews of PPI dispensing patterns and individualized feedback reports—will highlight deviations from evidence-based guidance, motivating pharmacists to reflect on and improve their performance. Simultaneously, establishing formal collaboration protocols between pharmacists and prescribers via shared electronic forms or dedicated hotlines will create a structured pathway for pharmacists to question inappropriate prescriptions and to suggest deprescribing when indicated [9,23,104]. Piloting “deprescribing rounds” in select districts can demonstrate the value of interdisciplinary dialogue and pave the way for broader adoption.

Looking further ahead, long-term, system-level interventions will embed PPI stewardship into the healthcare infrastructure. Multidisciplinary deprescribing teams [105] in major urban centers comprising pharmacists, physicians, and dietitians can be formed to oversee regional prescribing trends, develop tailored interventions, and serve as centers of excellence [23,106]. Parallel public education campaigns leveraging leaflets, social media, and community outreach will empower patients to ask informed questions about PPI necessity and to embrace lifestyle modifications that reduce dependence on acid-suppressing drugs [37]. Finally, integrating PPI case studies and deprescribing principles into undergraduate and postgraduate pharmacy curricula [22] will ensure that emerging pharmacists begin their careers with strong competencies in rational acid-suppressive therapy.

By front-loading targeted education and guideline access, followed by audit-driven feedback and collaborative protocols, and sustained by system reforms and public engagement, this prioritized framework offers a clear roadmap to optimizing PPI use, enhancing patient safety, and reducing unnecessary healthcare costs.

### 4.6. Strengths and Limitations

As far as we know, the published data on the KAPs of CPs regarding PPI use is sparse. This study is the first to evaluate CPs’ KAPs regarding PPI use. Our study has limitations. First, the cross-sectional, self-reported, online survey design and convenience sampling in two countries limit causal inference and generalizability and add selection bias. A future longitudinal/interventional study may help fill this gap and assess the impact of education, training, or policy change on PPI usage. Second, self-reported data may overestimate positive KAPs due to social desirability factors or add acquiescence bias. Third, a self-administered questionnaire prevented the research team from checking whether participants had consulted anyone or used references.

## 5. Conclusions

CPs have a legal role and an ethical obligation to educate patients about proper PPI usage and refer them in case of unexplained symptoms and adverse drug reactions. The current study identified the use of clinical guidelines as a predictor for the high knowledge scores of CPs regarding PPIs. CPs’ cumulative experience also was positively associated with their knowledge and practice results. These findings underscore the value of cooperation between healthcare authorities and the scientific societies for equipping CPs in Egypt and Iraq with updated knowledge and skills using continuing education programs and decision support systems regarding PPI use based on clinical guidelines. This can improve the knowledge, attitude, and practice gap. Encouraging physician–CP collaboration and patient participation in shared decision-making can help in deprescribing and the sensible usage of PPIs. Targeted education, using concise guideline summaries, case-based workshops, and decision-support tools, should focus on early-career pharmacists and areas with low guideline uptake. Future research should assess the impact of specific educational interventions and track longitudinal changes in CP behavior.

## Figures and Tables

**Figure 1 healthcare-13-01588-f001:**
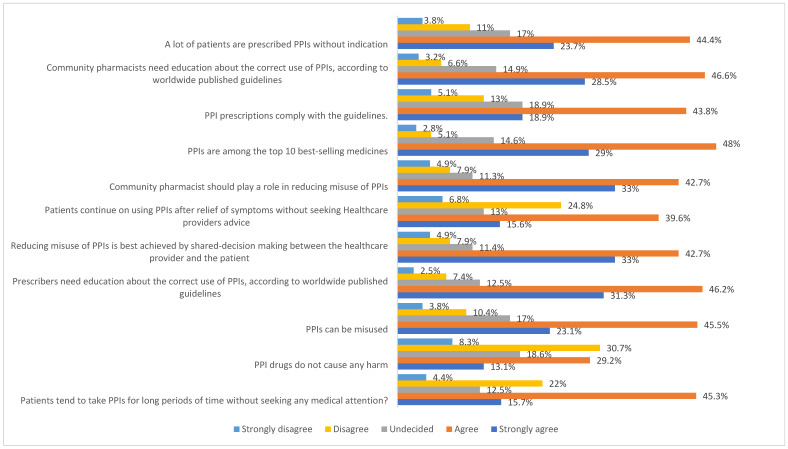
Attitudes of the study participants towards PPIs. PPIs: proton pump inhibitors.

**Figure 2 healthcare-13-01588-f002:**
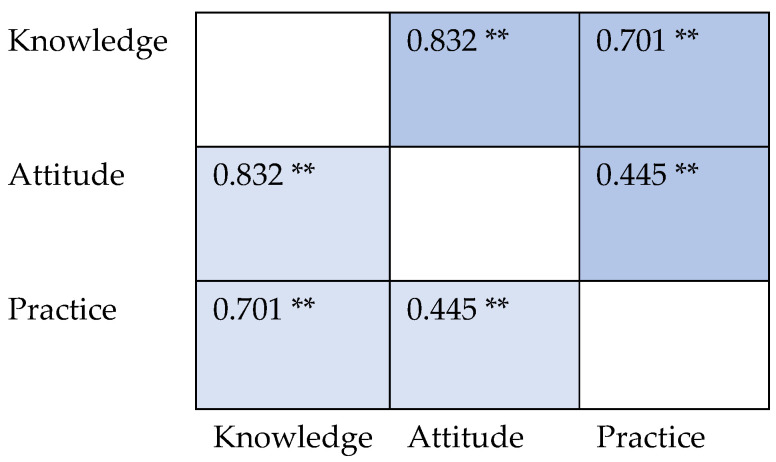
Spearman’s rank correlation between knowledge, attitude, and practice scores. ** means statistically significant (*p* value <0.05).

**Table 1 healthcare-13-01588-t001:** General characteristics of pharmacists involved in the study (N = 527).

Parameter	Total, n (%)	Knowledge Score (IQR)		Attitude Score, (IQR)		Practice Score, (IQR)	
**Sex**			*p* value		*p* value		*p* value
Female	252 (47.8%)	8 (6–9)	0.147	9 (6–11)	0.912	6 (4–8)	0.796
Male	275 (52.2%)	9 (8–12)		9 (6–9)		6 (3–12)	
**Years of experience**							
<1	106 (20.1%)	7 (6–7)	**0.001**	7 (5–9)	0.121	5 (4–7)	**0.021**
1–4	209 (39.7%)	7 (5–8)		7 (5–9)		6 (4–8)	
5–10	102 (19.4%)	8 (6–9)		9 (8–12)		7 (3–9)	
11–20	35 (6.6%)	7 (6–8)		9 (6–10)		7 (5–8)	
>20	75 (14.2%)	11 (6–14)		10 (8–12)		8 (4–12)	
**Highest degree**							
Bachelor of Pharmacy	368 (69.8%)	6 (4–8)	0.621	8 (7–9)	0.145	6 (4–7)	0.228
PharmD	56 (10.6%)	9 (6–12)		7 (6–12)		6 (4–6)	
Master’s	50 (9.5%)	8 (7–9)		10 (7–10)		5 (3–8)	
Doctorate (PhD)	53 (10.1%)	7 (6–15)		8 (6–8)		7 (3–11)	
**Country**							
Egypt	336 (63.8%)	9 (6–14)	0.897	9 (7–11)	0.089	6 (4–6)	0.689
Iraq	191 (36.2%)	9 (7–15)		10 (9–12)		6 (5–8)	
**Source of information about PPIs**							
Books	190 (36.1%)	7 (6–8)	**0.028**	8 (7–9)	0.105	6 (4–8)	0.239
Research articles	219 (41.6%)	9 (8–14)		9 (6–10)		5 (3–7)	
Colleagues	132 (25.0%)	8 (7–9)		7 (6–8)		6 (5–8)	
Facebook	214 (40.6%)	7 (6–11)		7 (6–9)		5 (3–9)	
Telegram	193 (36.6%)	9 (8–12)		8 (6–10)		7 (5–12)	
WhatsApp	174 (33.0%)	7 (6–8)		9 (7–10)		6 (5–10)	
Lexi comp	113 (21.4%)	7 (5–13)		9 (8–11)		6 (4–9)	
Drug Eye	190 (36.1%)	8 (6–8)		9 (8–11)		5 (4–8)	
GeneBrandex	51 (9.7%)	7 (6–9)		8 (7–9)		6 (4–9)	
Egyptian knowledge bank	110 (20.9%)	6 (5–8)		10 (7–12)		7 (5–9)	
Guidelines	135 (25.6%)	11 (7–16)		9 (8–11)		7 (5–9)	

The total median scores for knowledge, attitude, and practice were 11 out of 17 (IQR: 9−16), 9 out of 11 (IQR: 6−12), and 5 out of 6 (IQR: 3−8), respectively. Bold *p* values indicate significant results.

**Table 2 healthcare-13-01588-t002:** Knowledge of participants about PPIs.

Parameter	Total, n (%)
**Side effects caused by PPIs**	
Gastric carcinoids	215 (40.8%)
Hip fractures	179 (34.0%)
Hypomagnesemia	214 (40.6%)
Nutritional deficiencies	207 (39.3%)
Increased incidents of CVDs	85 (16.1%)
Enteric infections	141 (26.8%)
Diarrhea	146 (27.7%)
Community-acquired pneumonia	56 (10.6%)
Kidney diseases	77 (14.6%)
Dementia	102 (19.4%)
Change gut microbiota	106 (20.1%)
Duodenal G-cell tumors	81 (15.4%)
Anaphylaxis	42 (8.0%)
**Minerals and vitamins affected by PPIs**	
Calcium	270 (51.2%)
Magnesium	253 (48.0%)
Vitamin B12	342 (64.9%)
Manganese	106 (20.1%)
Potassium	118 (22.4%)
Sodium	91 (17.3%)
Selenium	48 (9.1%)
**Types of ulcers treated by PPIs**	
NSAIDs-induced ulcer	330 (62.6%)
*Helicobacter pylori*-induced ulcer	314 (59.6%)
Stress ulcer prophylaxis	210 (39.8%)
**Drugs interact with PPIs**	
Phenytoin	290 (55.0%)
Warfarin	278 (52.8%)
Clopidogrel	266 (50.5%)
Atazanvir	124 (23.5%)
Rilpivirine	140 (26.6%)
Nelfinavir	142 (26.9%)
Itraconazole	152 (28.8%)
Ketoconazole	176 (33.4%)
Posaconazole	86 (16.3%)
**What risk factors for ulcers and GI complications from NSAID use indicate the need for prophylactic PPIs?**	
Use of warfarin	194 (36.8%)
Use of anticoagulant	180 (34.2%)
Use of dexamethasone	176 (33.4%)
High-dose NSAIDs	239 (45.4%)
Longer duration of NSAIDs	231 (43.8%)
Low dose of aspirin	125 (23.7%)
**PPIs are clinically inferior to H2Ras, False**	254 (48.2%)
**Which of the following is correct?**	
All PPIs are OTC drugs.	197 (37.4%)
All PPIs are prescription-only medicine.	108 (20.5%)
Only some PPIs are OTC drugs.	222 (42.1%)
**The administration of PPIs with ticlopidine or clopidogrel or anti-coagulants alone without risk factors is recommended, False**	221 (41.9%)
**In patients taking steroids alone for whatever clinical condition, mucosal protection with a PPI is routinely indicated, False**	142 (26.9%)
**Sudden withdrawal of PPIs is not recommended, True**	300 (56.9%)
**For which of the following categories of patients using NSAIDs and with no other risk factors are PPIs indicated for gastroprotection?**	
45–55 years	214 (40.6%)
56–65 years	119 (22.6%)
>65 years	194 (36.8%)
**Esomeprazole, lansoprazole, and omeprazole work best when taken:**	
30 min before breakfast	365 (69.3%)
After food	130 (24.7%)
With food	32 (6.1%)
**What is the duration PPIs could be safely used without referring to a specialized physician?**	
2 weeks	278 (52.8%)
2 months	107 (20.3%)
3 months	57 (10.8%)
Indefinitely	85 (16.1%)
**In case of persistent and severe night symptoms, it is recommended to:**	
Take PPIs in the morning	176 (33.4%)
Take PPIs before dinner	180 (34.2%)
Fraction the daily dose into two separate administrations, one before breakfast and the other before dinner	171 (32.4%)
**PPI therapy should be prescribed to treat chronic laryngitis, False**	241 (45.7%)
**PPIs can improve outcomes in Barrett’s esophagus, True**	311 (59.0%)
**Like H2RA, PPIs can cause rapidly decreasing response to the drug (tachyphylaxis), False**	251 (47.6%)

CVDs: cardiovascular diseases; GI: gastrointestinal; H2RAs: H2-receptor antagonists; NSAIDs: non-steroidal anti-inflammatory drugs; OTC: over the counter; PPIs: proton pump inhibitors.

**Table 3 healthcare-13-01588-t003:** The practice of participants towards PPIs (N = 527).

Parameter	Total, n (%)
**Provide advice for patients who use PPIs about lifestyle changes to alleviate their symptoms**	
Always	263 (49.9%)
Often	125 (23.7%)
Sometimes	90 (17.1%)
Rarely	37 (7.0%)
Never	12 (2.3%)
**Contact the prescriber or advise the patient to stop PPIs when there is no current indication for their use**	
Always	164 (31.1%)
Often	172 (32.6%)
Sometimes	126 (23.9%)
Rarely	46 (8.7%)
Never	19 (3.6%)
**Report ADR of PPIs to the manufacturer or regulatory authorities**	
Always	102 (19.4%)
Often	127 (24.1%)
Sometimes	129 (24.5%)
Rarely	80 (15.2%)
Never	89 (16.9%)
**Consider PPIs the first choice when recommending acid-suppression drugs**	
Always	143 (27.1%)
Often	154 (29.2%)
Sometimes	148 (28.1%)
Rarely	45 (7.0%)
Never	37 (7.0%)
**Prescribe PPI for patients, Yes**	450 (85.4%)
**Use guidelines such as the JSGE or ACG when prescribing PPIs, Yes**	396 (75.1%)

ADR: adverse drug reactions; ACG: American College of Gastroenterology; JSGE: Japanese Society of Gastroenterology; PPIs: proton pump inhibitors.

## Data Availability

Data are available from the corresponding author (H.A.A.M) upon reasonable request.

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
