# Peer review of "Knowledge, Attitudes, and Practices of Community Pharmacists Regarding Proton Pump Inhibitor (PPI) Use: A Cross-Sectional Study"

_healthcare, 2025, doi:10.3390/healthcare13131588_

Round 1
Reviewer 1 Report (Previous Reviewer 4)
Comments and Suggestions for Authors
I read with interest the paper titled "Knowledge, Attitudes, and Practices of Community Pharmacists Regarding Proton Pump Inhibitors (PPIs) Use: A Cross-Sectional Study"
1. Objective of the study should be highlighted in the abstract.
2. What's the rational behind the the choose of the countries involved in the study?
3. Any validation of the questionnaire, apart of the testing in the 30 volunteers? Are those 30 CPs results added to the final sample?
4. How do you assess 15000 CPs per country? Any available data or just the perception of the authors? Seems strange a very round number on that. Moreover, accordingly to the data provided (15000 CPs per country with 95 % CI and 5 % margin),calculation of sample size is either overestimated (if you use 15000x2 as final population) or underestimated (if you use independent sample sizes). Please check the calculation and provide evidence on that.
4.1. In the abstract author state "A sample size of 525 was based on an estimated 15000 CPs per country (95 % CI, 5 % margin)." - in the main methods, different information of 385 was presented. This is contradictory, however 385 is correct.
5. In the abstract "The total median scores for knowledge, attitude, and practice were 11 out of 17 (IQR: 9-16), 9 out of 11 (IQR: 6-12), and 5 out of 6 (IQR: 3-8), respectively." - this is for Egypt or Iraq or overall?
Author Response
Comment 1:
I read with interest the paper titled "Knowledge, Attitudes, and Practices of Community Pharmacists Regarding Proton Pump Inhibitors (PPIs) Use: A Cross-Sectional Study"
- Objective of the study should be highlighted in the abstract
Response 1: Thank you so much for your effort provided to improve our manuscript. We emphasized the objective of the study in the abstract as recommended.
Comment 2:
What's the rational behind the the choose of the countries involved in the study?
Response 2:Thank you for this key comment. First, we chose these countries due to conveinience as some of the authors currently work there. Second, there is gap in literature about the KAPs of PPIs of community pharmacists in Egypt and Iraq which encouraged us to choose our topic of research.
Comment 3:
Any validation of the questionnaire, apart of the testing in the 30 volunteers? Are those 30 CPs results added to the final sample?
Response 3:Thank you for you question. The items of the questionnaire was extracted from validated questionnaire and we did not conduct a new validation, instead, we tested them against 30 volunteers who were not included in the final sample data analysis.
Comment 4:
How do you assess 15000 CPs per country? Any available data or just the perception of the authors? Seems strange a very round number on that. Moreover, accordingly to the data provided (15000 CPs per country with 95 % CI and 5 % margin),calculation of sample size is either overestimated (if you use 15000x2 as final population) or underestimated (if you use independent sample sizes). Please check the calculation and provide evidence on that.
Response 4:
Thank you for this remark. we have now replaced it with the latest registry counts for Egypt (75 165 community pharmacies) and Iraq (22 120 registered pharmacists) and recalculated the sample-size justification accordingly.;
“A sample size of 385 CPs was required based on an estimated 93 000 community pharmacists in Egypt and and 22 120 in Iraq; however, to improve statstical power, we aimed to include >500 CPs.”
Comment 4.1. In the abstract author state "A sample size of 525 was based on an estimated 15000 CPs per country (95 % CI, 5% margin)." - in the main methods, different information of 385 was presented. This is contradictory, however 385 iscorrect. The sample size must be clarified and be consistent.
Response 4.1 Thank you for your feedback. Please be informed that we corrected any mistakes in writing and we would like to clarify that the the target population was estimated at roughly 93,000 community pharmacists in Egypt and 22,120 in Iraq. Using the Raosoft cal-culator with this population frame, a 95 % confidence level and a 5 % margin of error indicated a minimum sample of 384 re-spondents; to enhance statistical power and representativeness we therefore aimed to obtain—and ultimately exceeded—500 com-pleted questionnaires. We made this unified in both abstract
Comment 5:
In the abstract author state "A sample size of 525 was based on an estimated 15000 CPs per country (95 % CI, 5 % margin)." - in the main methods, different information of 385 was presented. This is contradictory, however 385 is correct.
Response 5:
Thank you. We agree with you. The Methods section is correct, indicating a minimum calculated sample of 384 (rounded to 385) and 525 completed questionnaires analysed. We have revised the abstract to mirror this wording so it now states that 384/385 was the required minimum and 525 the final sample size. To improve statistical power, we targeted ≥ 500 responses and ultimately analysed 525 fully completed questionnaires.
Comment 6:
In the abstract "The total median scores for knowledge, attitude, and practice were 11 out of 17 (IQR: 9-16), 9 out of 11 (IQR: 6-12), and 5 out of 6 (IQR: 3-8), respectively." - this is for Egypt or Iraq or overall?
Response 6:Thanks, those are overall median scores of KAPs
Reviewer 2 Report (Previous Reviewer 3)
Comments and Suggestions for Authors
The manuscript presents a well-structured cross-sectional study assessing community pharmacists’ knowledge, attitudes, and practices (KAP) regarding PPI use in Egypt and Iraq. It fills an important gap, as misuse of PPIs and the role of community pharmacists in regulating their use has received limited attention in the Middle East.
The recommendations are good, but could benefit from prioritization (e.g., continuing education modules vs. national guideline dissemination).
Author Response
comment 1:
The manuscript presents a well-structured cross-sectional study assessing community pharmacists’ knowledge, attitudes, and practices (KAP) regarding PPI use in Egypt and Iraq. It fills an important gap, as misuse of PPIs and the role of community pharmacists in regulating their use has received limited attention in the Middle East.
Response 1:Thank you so much for your time and effort exerted to enhance the quality of our manuscript.
Comment 2:The recommendations are good, but could benefit from prioritization (e.g., continuing education modules vs. national guideline dissemination).
Response 2:In accordance with the suggestion, we updated the recommendation sections as recommended
Reviewer 3 Report (Previous Reviewer 2)
Comments and Suggestions for Authors
The article is long. To increase the percentage of readers who read the article, its size should be reduced.
Author Response
Comment 1:
The article is long. To increase the percentage of readers who read the article, its size should be reduced.
Response 1:We appreciate the reviewer’s feedback. We tried to reduce the word count of the manuscript to make it more condensed and to the point
This manuscript is a resubmission of an earlier submission. The following is a list of the peer review reports and author responses from that submission.
Round 1
Reviewer 1 Report
Comments and Suggestions for Authors
Thank you for the opportunity to review this manuscript titled “Knowledge, Attitudes, and Practices of Community Pharmacists Regarding Proton Pump Inhibitor (PPI) Use: A Cross-Sectional Study”. The topic is relevant, however, several major concerns must be addressed before the manuscript can be considered for publication.
1. Please consider professional English language editing to improve the manuscript.
2. No clear description about how the questionnaire was designed. Is it adapted from validated tools? Please describe how the items were chosen development process, the response formats, and the scoring.
3. Please explain how responses in each domain (knowledge, attitude, practice) were converted into scores.
4. Please define the thresholds used to classify respondents’ scores into acceptable/satisfactory or not
6. Cronbach’s alpha was only reported overall. Please also report subscale alpha values for each domain (knowledge, attitude, practice) to demonstrate internal consistency more transparently.
7. Please include the response rate
8. Consider adding a brief statement clarifying how data security and anonymity were maintained, particularly in the context of online data collection.
9. The manuscript claims that CPs had "satisfactory" attitudes and practices, yet the result of 'only 19.4% always reported ADRs' suggest suboptimal behavior. Please clarify this interpretation.
10. The correlation coefficient of 0.445 between attitude and practice is only moderate. Please avoid overstating and interpret accordingly.
11. The item “PPIs are among the top 10 best-selling medicines” appears in the attitude domain. Please justify its inclusion in this domain.
12. You report that pharmacists who used clinical guidelines scored higher. Please elaborate: How did source of information influence knowledge? Why certain sources significantly associated with poor knowledge?
13. The abstract lacks specific numeric results for KAP scores. Please include at least the median scores.
14. Please include the full KAP questionnaire used in this study as a Supplementary Appendix
Please consider professional English language editing to improve the manuscript.
Reviewer 2 Report
Comments and Suggestions for Authors
Please find the attached file.

Reviewer 3 Report
Comments and Suggestions for Authors
- The manuscript addresses an important public health and clinical pharmacy issue, given the widespread use/misuse of PPIs and the evolving role of CPs in fostering rational drug use. Although the manuscript states that a convenience sampling method was used (recruiting CPs via visits and social media groups), it would be stronger if the authors provided more details on:
- How the pharmacies and CPs were selected.
- The response rate and any nonresponse bias.
- Is the sample representative of CPs in both countries, especially given potential cultural and systemic differences between Egypt and Iraq?
- The study uses a cross-sectional design, which is suitable; however, the authors need to recognize that associations (e.g., between years of experience and knowledge scores) do not establish causality.
3. Questionnaire Design:
The instrument addresses various domains (knowledge, attitudes, and practices) and was reportedly validated by experts (content validity index of 8.1) and pilot-tested with 16 CPs. Nonetheless, additional details are necessary on:
- How questions were constructed (e.g., are the knowledge items based on current clinical guidelines?).
- The process of determining “correct” answers, especially for items where expert consensus might vary.
- The manuscript notes that only a small percentage of CPs recognized certain PPI side effects (e.g., pneumonia, anaphylaxis). The authors should discuss possible reasons for these gaps and the implications for clinical practice.
- Comparisons with similar studies (both regionally and internationally) would enrich the discussion. What might explain differences in knowledge or practice patterns? How do local policies, education systems, or drug availability play a role?
While some limitations (e.g., reliance on self-reported data) are noted, the authors should more fully discuss
- The potential for selection bias due to convenience sampling or social desirability.
- The possibility of social desirability bias in responses—CPs may over-report “good practice” behaviors.
- How cross-sectional design limits the understanding of changes over time.
- Recommendations for Future Research
A discussion of how longitudinal studies or interventional trials might further elucidate the impact of educational interventions on CPs’ KAP regarding PPIs would be a useful addition.
7. Ensure all abbreviations (e.g., CPs, PPIs, NSAIDs, H2RAs) are defined at first appearance.
Comments on the Quality of English LanguageMinor grammatical and typographical errors need correction.
Reviewer 4 Report
Comments and Suggestions for Authors
I read with interest the paper titled "Knowledge, Attitudes, and Practices of Community Pharmacists Regarding Proton Pump Inhibitors (PPIs) Use: A Cross-Sectional Study".
1. Gender should be sex instead.
2. What tests are being used in table 1 that present p-values? They are not described in the methods section.
3. Sample size should have been calculated per country (as a subgroup) and not overall. This is a limitation.
4. What is a difference between a bachelor in pharmacy and a PharmD? Are ones pharmacy technicians and another ones pharmacists?
5. Figure 1 will gain with the percentages (relative frequency) presented.
6. Table 2 present knowledge. How was that knowledge assessed? In yes/no question or tick box questions? I cannot understand from the survey added in the end? Despite of this, questions formulated like this, often leads to social disarability bias, knowledge bias and acquiscence bias, that should be further discussed in the limitations.
7. I expected correlations between knowledge, attitudes and practices and the agreement with the formulated questions. Authors only shown the correlation between the three variables, which doesnt add much about the topic and the objective of the manuscript.